# Endoscopic papillary balloon dilatation less than three minutes for biliary stone removal increases the risk of post-ERCP pancreatitis

**Chung-Kai Chou**[1,2,3]**, Kuei-Chuan Lee**[1,2]*, **Jiing-Chyuan Luo**[1,2]*, **Tseng-Shing Chen**[1,2]**, Chin-Lin Perng**[1,2]**, Yi-Hsiang Huang**[1,2]**, Han-Chien Lin**[1,2]**, Ming-Chih Hou**[1,2]

**1** Division of Gastroenterology and Hepatology, Department of Medicine, Taipei Veterans' General Hospital, Taipei, Taiwan, **2** Department of Medicine, National Yang-Ming University School of Medicine, Taipei, Taiwan, **3** Division of Gastroenterology, Department of Medicine, National Yang-Ming University Hospital, Ilan, Taiwan

* kclee2@vghtpe.gov.tw (KCL); jcluo@vghtpe.gov.tw (JCL)

## Abstract

### Objectives

The adequate duration for EPBD was unclear. Therefore, we aimed to investigate the effect of balloon dilatation duration of EPBD on the occurrence of PEP.

### Methods

One hundred and ninety-eight patients with common bile duct (CBD) stone treated by EPBD were retrospectively recruited. The dilatation duration was determined according to adequate opening of the biliary orifice without bleeding. The clinical outcomes and complications of EPBD were recorded.

### Results

We stratified the patients according to dilatation duration (Group A, <3 minutes; Group B, 3–5 minutes; Group C, $\geq$5 minutes). The group C patients had a higher proportion of large CBD stones (stones $\geq$10 mm) (33.3% vs. 26.8% vs. 53.5%, p = 0.01). Patients in group A had a significantly higher PEP rate than patients in group B (13.3 vs. 3.1, p = 0.032). There were no significant differences in perforation and bleeding rate among the three groups. Univariate and multivariate analyses showed that a dilatation duration of <3 minutes, CBD diameter < 10 mm and age $\leq$ 75 years were independent risk factors of PEP in post-EPBD patients.

### Conclusions

In patients receiving EPBD, dilatation duration <3 minutes, lower CBD diameter, and younger age were independent risk factors of PEP.

**Data Availability Statement:** All relevant data are within the manuscript.

**Funding:** The study was partly supported by grants from Ministry of Science and Technology of Taiwan

(108-2628-B-075-008) and Taipei Veterans General Hospital (V109C-118).

**Competing interests:** The authors have declared that no competing interests exist.

**Abbreviations:** EPBD, Endoscopic papillary balloon dilatation; PEP, post endoscopic retrograde cholangiopancreatography pancreatitis; ERCP, endoscopic retrograde cholangiopancreatography; CBD, common bile duct; EST, endoscopic sphincterotomy; EML, endoscopic mechanical lithotripsy; CT, computed tomography; MRCP, magnetic resonance cholangiopancreatography.

## Introduction

Choledocholithiasis, or common bile duct (CBD) stones, is a common disease globally. Endoscopic retrograde cholangiopancreatography (ERCP) with biliary endoscopic sphincterotomy (EST) has become the preferred therapeutic procedure for CBD stone removal. Endoscopic papillary balloon dilation (EPBD) was developed as a comparable alternative method to EST since the 1990s [1, 2]. Because of a lower risk of bleeding and preservation of papillary function [3], EPBD is comparable to EST for stone extraction, though it may require more endoscopic mechanical lithotripsy (EML) [4–6]. Furthermore, EPBD is usually preferred in younger patients or patients with coagulopathy [7, 8].

However, the biggest concern of EPBD is the increased rate of post-ERCP pancreatitis (PEP) [4, 5, 9, 10]. Some studies have found that the duration of EPBD was inversely associated with the risk of PEP [11]. However, the adequate dilatation time is still unclear. A randomized trial by Liao et al. found that the risk of PEP was lower with a 5-minute EPBD than with a 1-minute EPBD (4.8% vs. 15.1%), with a relative risk of 0.32 [12]. Chan et al. performed large balloon dilatations alone without sphincterotomy in 247 patients with large CBD stones. The mean duration of the dilatation procedure was 4.7 minutes, and the PEP rate was 0.8% (2/247) [13]. Oh et al. compared endoscopic papillary large balloon dilatation (EPLBD) and EST for removal of large CBD stones, with a duration of balloon dilatation of 31.3 seconds and a PEP rate of 5.0% [14]. Additionally, a meta-analysis showed that short EPBD (≤1 minute) had a higher risk of pancreatitis (odds ratio, 3.87) [11]. Currently, it is understood that balloon dilatation ≤1 minute actually increases the risk of pancreatitis in patients with CBD stones less than 1 cm [11, 12]. A compartment syndrome from post-EPBD hemorrhage/edema with uncut Sphincter of Oddi has been the proposed mechanism for the increased risk of pancreatitis [12]. However, not every patient can tolerate dilatation-induced pain for more than 5 minutes. In our hospital, the EPBD or EPLBD (balloon size >10mm) was stopped when an adequate orifice was opened without active bleeding. In addition, in patients receiving EPLBD, there have been no studies investigating the association of balloon dilation time with PEP. Therefore, in this study, we aimed to investigate the effect of the balloon dilation time on the efficacy and complications of EPBD or EPLBD in the removal of bile duct stones.

## Methods

### Patients

Between November 2010 and October 2012, 198 patients who were admitted to Taipei Veterans General Hospital due to CBD stones and treated by EPBD/EPLBD were retrospectively reviewed. The diagnosis of CBD stone was made by an abdominal sonogram, computed tomography (CT) scan, or magnetic resonance cholangiopancreatography (MRCP). The Ethical Review Committee of the Taipei Veterans General Hospital approved this study.

### Endoscopic papillary balloon dilation

The EPBD procedure was performed by three experienced endoscopists (KCL, JCL, and TSC, > 200 procedures/year) via a duodenoscope (Olympus, JF260V). The experienced endoscopist evaluated the papilla first. In general, we used a cannula (Olympus, StarTipV, PR-V418Q) first (n = 152). If a small orifice of the papilla was noted, we used a taper cannula (Olympus, StarTipV, PR-V220Q) (n = 46). If the cannula and taper cannula both failed, we tried a guidewire-assisted cannulation (Olympus, VisiGlide2, G-26—2545S). None of these patients received precuts due to difficult cannulations. A difficult cannulation was defined after 5 minutes or five attempts, or more than one pancreatic cannulation (n = 54, 27.3%) [15]. We used CRE™

Balloon Dilatation Catheters (Boston scientific) for papillary dilatation, the balloon sizes ranged 8-15mm. The termination of the dilatation was decided subjectively according to an adequate biliary orifice opened without bleeding, and the dilatation time was recorded by a timer. After papillary dilatation, the CBD stone was extracted by the single-use retrieval basket (Olympus, TetraCatchV, FG-V422PR) or the balloon extractor (CONMED, DURAglide3 stone balloon) with/without lithotripsy (Olympus, LithoCrushV and lithotripsy handle MAJ-441).

### Data collection

Medical records were reviewed for data collection. The following data were collected: (1) patients characteristics, such as age, sex, underlying diseases, and previous abdominal surgery history; (2) laboratory data, such as complete blood cell counts, international normalized ratio (INR), and serum levels of alanine aminotransferase (ALT), aspartate aminotransferase (AST), total bilirubin (TB), alkaline phosphatase (Alk-P), γ-glutamyl-transferase (γ-GT), blood urea nitrogen (BUN), creatinine, measured by Rochi/Hitachi Modular Analytics Systems (Roche Diagnostics GmbH, Mannheim, Germany); (3) findings during the procedure, such as maximum CBD diameter, CBD stone size, pancreatic duct cannulation, balloon dilatation time, and the tool used for the stone removal (basket or extraction balloon); (4) procedure-related complications and mortality; and (5) in hospital mortality and causes of mortality.

### Statistical analysis

All statistical analyses were performed using the SPSS 17.0 for Windows (SPSS. Inc., Chicago, IL, USA). Continuous variables were presented as mean ± standard deviation. Logistic regression was used to determine the effects of variables on post ERCP pancreatitis. Only variables with p-values proximal to 0.1 in the univariate analysis were selected for the multivariate analysis. The effects of categorical variables on the outcomes were tested by the chi-square test. A two-tailed p value $< 0.05$ was considered to be significant.

## Results

### Basic characteristics

In total, 198 patients with CBD stones were admitted to our hospital and treated by EPBD. The mean age was 71.2±15.0 years (range, 25–97 years), and 128 (64.6%) patients were male. We separated the patients into three groups according to dilatation duration (t, minutes) (Group A, t< 3; Group B, 3≤ t < 5; Group C: t ≥ 5).

The basic characteristics of the three groups are shown in Table 1. There was no significant difference in age, sex, body mass index, and underlying comorbidities. The clinical conditions before EPBD- including vital signs, blood cell counts, renal function, liver function, and biliary tract profile were not significantly different.

### Dilatation time was associated with CBD stone size

The findings during the EPBD procedure are shown in Table 2. There were no significant differences in the maximum CBD diameter, proportion of juxtapapillary diverticulum, and lithotripsy used between the three groups. However, patients in Group C had a larger CBD stone size (9.1±4.2 vs. 8.2±3.8 vs. 11.0±5.7mm, p = 0.003) and higher proportion of larger CBD stone (stones ≥ 10mm: 33.3 vs. 26.8 vs. 53.5%, p = 0.01). These results imply that bigger CBD stones need more dilatation time for EPBD to stop bleeding and ensure orifice opening.

**Table 1. Basic characteristics of the patients.**

| Group | A (t< 3) | B (3≦ t < 5) | C (t ≧ 5) | P |
|---|---|---|---|---|
| N (number) | 30 | 97 | 71 | |
| Age (year-old) | 69.37±15.3 | 69.6±15.8 | 74.3±13.4 | .105 |
| Sex (M) | 17 (56.7) | 68 (70.1) | 43 (60.6) | .270 |
| BMI | 24.9±3.9 | 23.3±3.5 | 23.9±4.0 | .176 |
| BW (kg) | 62.8±13.0 | 61.8±10.7 | 62.6±12.2 | .869 |
| Comorbidity | | | | |
| CV | 17 (56.7) | 49 (50.5) | 48 (67.6) | .086 |
| CM | 4 (13.3) | 8 (8.2) | 2 (2.8) | .139 |
| Neurology | 6 (20.0) | 13 (13.4) | 15 (21.1) | .383 |
| DM | 8 (26.7) | 16 (16.5) | 17 (23.9) | .341 |
| Nephrology | 3 (10.0) | 6 (6.2) | 5 (7.0) | .776 |
| Malignancy | 3 (10.0) | 10 (10.3) | 13 (18.3) | .272 |
| SBP (mmHg) | 139.5±20.2 | 131.3±26.4 | 139.2±29.5 | .114 |
| BT (˚C) | 36.8±0.8 | 37.0±1.1 | 37.0±1.2 | .806 |
| HR (/min) | 86.5±18.7 | 82.5±19.9 | 84.4±18.3 | .588 |
| RR (/min) | 19.1±2.2 | 19.0±1.7 | 19.8±6.8 | .481 |
| WBC (/cumm) | 9462.3±4561.7 | 10634.4±6290.4 | 9522.8±5135.6 | .376 |
| Hb (g/dL) | 12.6±2.2 | 12.9±1.5 | 12.7±1.8 | .672 |
| PLT (/cumm) | 219285.7±111629.6 | 203684.2±71627.2 | 189676.1±77174.0 | .230 |
| INR | 1.1±0.1 | 1.1±0.1 | 1.1±0.1 | .629 |
| ALT (U/L) | 170.0±302.4 | 228.1±229.4 | 173.5±174.3 | .235 |
| AST (U/L) | 211.9±580.3 | 251.1±268.6 | 229.5±263.2 | .855 |
| BUN (mg/dL) | 18.7±9.1 | 17.6±8.2 | 19.0±11.4 | .671 |
| Cr (mg/dL) | 1.1±0.6 | 1.2±0.6 | 1.3±1.6 | .587 |
| ALP (U/L) | 212.1±172.3 | 224.7±149.3 | 226.8±174.5 | .919 |
| rGT (U/L) | 321.6±340.9 | 373.2±298.8 | 331.6±355.6 | .653 |
| Amy (U/L) | 655.9±953.4 | 570.0±955.0 | 993.6±1538.4 | .372 |
| Tbili (mg/dL) | 3.2±2.7 | 3.6±3.8 | 3.1±2.7 | .617 |
| CRP (mg/dL) | 7.9±8.6 | 6.0±6.6 | 5.2±5.6 | .267 |
| Antiplatelet | 4 (13.3) | 13 (13.4) | 10 (14.1) | .991 |
| Anticoagulant | 1 (3.3) | 0 (0) | 2 (2.8) | .227 |
| Bile culture | 1 (3.3) | 12(12.4) | 9 (12.7) | .338 |
| Blood culture | 2 (6.7) | 16(16.5) | 8 (11.3) | .320 |
| PTCD/PTGBD | 1(3.3) | 9 (9.3) | 7 (9.9) | .389 |
| Previous Abdominal Surgery | 8 (26.7) | 28 (28.9) | 24 (33.8) | .707 |
| Previous EST | 0 (0) | 4 (4.1) | 4 (5.6) | .421 |
| Previous EPBD | 0 (0) | 1 (1.0) | 3 (4.2) | .241 |

t: endoscopic papillary balloon dilatation duration time, N: number, BMI: body mass index, BW: body weight, CV: cardiovascular, CM: chest, DM: diabetes mellitus, SBP: systolic blood pressure, BT: body temperature, HR: heart rate, RR: respiratory rate, WBC: white blood cell count, Hb: hemoglobin, PLT: platelet, INR: international normalized ratio, ALT: alanine aminotransferase, AST: aspartate aminotransferase, BUN: blood urea nitrogen, Cr: creatinine, ALP: alkaline phosphatase, rGT: gamma-glutamyl transpeptidase, Amy: amylase, Tbili: total bilirubin, CRP: c-reactive protein, BD: bile duct, CBD: common bile duct, PTCD: percutaneous transhepatic cholangiography and drainage, PTGBD: percutaneous transhepatic gallbladder drainage, EST: endoscopic sphincterotomy, EPBD: endoscopic papillary balloon dilatation.

**Table 2. Findings during procedure.**

| Group | A (t< 3) | B (3≦ t < 5) | C (t ≧ 5) | P |
|---|---|---|---|---|
| N (number) | 30 | 97 | 71 | |
| CBD stone > = 10mm | 10 (33.3) | 26 (26.8) | 38 (53.5) | .010 |
| Maximum CBD diameter (mm) | 15.1±5.0 | 14.8±4.8 | 16.2±5.1 | .175 |
| CBD stone size (mm) | 9.1±4.2 | 8.2±3.8 | 11.0±5.7 | .003 |
| JPD | 10 (33.3) | 41 (42.3) | 35 (49.3) | .318 |
| Pus | 2(6.7) | 3 (3.1) | 4(5.6) | .632 |
| Sludge | 5 (16.7) | 26(26.8) | 19(26.8) | .481 |
| Balloon size (mm) | | | | .108 |
| 8 | 11 | 29 | 12 | |
| 10 | 15 | 46 | 32 | |
| 11 | 0 | 3 | 4 | |
| 12 | 2 | 17 | 16 | |
| 14 | 1 | 2 | 6 | |
| 15 | 1 | 0 | 1 | |
| EPLBD (>10mm) | 4 (13.3%) | 22 (22.7%) | 27 (38%) | .017 |
| Complete removal | 30 (100%) | 97 (100%) | 71 (100%) | |
| Extraction Balloon | 10(33.3) | 36(37.1) | 24(33.8) | .876 |
| Retrieval Basket | 15 (50.0) | 56 (57.7) | 47 (66.2) | .354 |
| Precut | 0 (0) | 0 (0) | 0(0) | - |
| Schendra | 0 (0) | 0 (0) | 0(0) | - |
| Lithotripsy | 0(0) | 2 (2.1) | 5 (7.0) | .118 |

CBD: common bile duct, JPD: juxtapapillary diverticulum, EPLBD: endoscopic papillary large balloon dilatation.

## EPBD complications

Table 3 shows the complications after EPBD. Overall, there were no significant differences in the PEP rate between the three groups (13.3% vs. 3.1% vs. 4.2%, p = 0.075). However, the subgroup analysis showed that the group A patients had a significant higher PEP rate than the group B patients (13.3% vs. 3.1%, p = 0.032). There were no perforations or deaths that occurred in the three groups. The bleeding rate, post EPBD biliary tract infection, and aspiration pneumonia were not significantly different. In addition, there was no difference in biliary tract infection when compared the Group A (n = 0) with the Groups B and C together (n = 10) (p = 0.170).

**Table 3. Post endoscopic papillary balloon dilatation complication.**

| Group | A (t< 3) | B (3≦ t < 5) | C (t ≧ 5) | P |
|---|---|---|---|---|
| N (number) | 30 | 97 | 71 | |
| Bleeding | 0 (0) | 2 (2.1) | 0 (0) | .349 |
| Perforation | 0 (0) | 0(0) | 0(0) | - |
| Pancreatitis | 4(13.3)* | 3(3.1)* | 3(4.2) | .075 |
| BTI | 0(0) | 4 (4.1) | 6 (8.5) | .175 |
| Aspiration Pneumonia | 0 (0) | 0 (0) | 1 (1.4) | .407 |
| In hospital death | 0(0) | 0 (0) | 0 (0) | - |

* P = 0.032 between Group A and B.

BTI: biliary tract infection.

**Table 4. Univariate and multivariate analysis of post ERCP pancreatitis.**

| | Univariate Analysis | | | Multivariate Analysis | | |
|---|---|---|---|---|---|---|
| | PEP (+) n = 10 | PEP (-) n = 188 | P value | P value | OR | 95% CI |
| Age (≤75-year-old) | 8/10 (80.0%) | 91/188 (48.4%) | **.052** | **.057** | 5.006 | 0.954–26.286 |
| Sex (male) | 4/10 (40.0%) | 124/188 (66.0%) | **.094** | NA | | |
| Total bilirubin (≤ 1.2 mg/dL) | 5/10 (50.0%) | 48/188 (25.5%) | **.096** | NA | | |
| CBD < 10mm | 3/10 (30.0%) | 19/188 (10.1%) | **.051** | **.034** | 5.332 | 1.134–25.058 |
| Pancreatic duct cannulation | 3/10 (30.0%) | 36/188 (19.1%) | .401 | NA | | |
| JPD (+) | 3/10 (30.0%) | 83/188 (44.1%) | .379 | NA | | |
| Balloon size | | | .577 | | | |
| EPLBD (>10mm) | 1/10 (10.0%) | 52/188 (27.7%) | .267 | NA | | |
| Difficult cannulation* | 5/10 (50.0%) | 49/188 (26.1%) | **.098** | NA | | |
| CBD stone < 10mm | 7/10 (70.0%) | 92/188 (48.9%) | .201 | NA | | |
| Lithotripsy (+) | 0/10 (0.0%) | 7/188 (3.7%) | .534 | | | |
| Dilatation < 3min | 4/10 (40.0%) | 26/188 (13.9%) | **.049** | **.027** | 4.942 | 1.194–20.447 |
| Dilatation < 5min | 7/10 (70.0%) | 120/188 (63.8%) | .692 | NA | | |
| Time | | | .332 | NA | | |

ERCP: endoscopic retrograde cholangiopancreatography, PEP: post ERCP pancreatitis, OR: odds ratio, CI: Confidence interval, NA: not available, NS: not significant,

CBD: common bile duct, JPD: juxtapapillary diverticulum, EPLBD: endoscopic papillary large balloon dilatation

*: cannulation attempts of duration > 5 minutes, > 5 attempts, or 2 pancreatic guidewire passages.

### Post-ERCP pancreatitis rate

We further analyzed the risk factor of post-EPBD pancreatitis (Table 4). In the univariate analysis, an age ≤ 75 years (p = 0.052), female (p = 0.094), total bilirubin ≤1.2 mg/dL (p = 0.096), CBD diameter < 10 mm (p = 0.051), difficult cannulation (p = 0.098), and dilatation duration < 3minutes (p = 0.049) were possible risk factors of PEP. A further multivariate analysis showed that age ≤ 75 years (OR, 5.006; p = 0.057), CBD < 10 mm (OR, 5.332; p = 0.034), and dilatation duration < 3 minutes (OR, 4.942; p = 0.027) were still independent risk factors of post-EPBD pancreatitis. In 53 patients who received EPLBD, the dilatation time for a minimum of three minutes did not increase the PEP rate (0% vs. 4.5% vs. 0%, p = 0.488) (Table 5).

## Discussion

The current consensus suggests EPBD as an alternative to EST for extracting CBD stones, especially in the presence of coagulopathy or altered anatomy [7]. However, the biggest

**Table 5. Post endoscopic papillary balloon dilatation complication in EPLBD patients.**

| Group | A (t< 3) | B (3≦ t < 5) | C (t ≧ 5) | P |
|---|---|---|---|---|
| N (number) | 4 | 22 | 27 | .017 |
| Bleeding | 0 (0) | 1 (4.5) | 0 (0) | .488 |
| Perforation | 0 (0) | 0 (0) | 0 (0) | - |
| Pancreatitis | 0 (0) | 1 (4.5) | 0 (0) | .488 |
| BTI | 0 (0) | 2 (9.1) | 1 (3.7) | .632 |
| Aspiration Pneumonia | 0 (0) | 0 (0) | 0 (0) | - |
| In hospital death | 0 (0) | 0 (0) | 0 (0) | - |

EPLBD: Endoscopic papillary large balloon dilatation; BTI: Biliary tract infection.

concern of EPBD was the increased rate of post-EPBD pancreatitis [2, 4, 5, 9, 10, 16]. Previous studies have demonstrated that the incidence of post-EPBD pancreatitis was 7.4–15.4% [2, 4, 5, 9]. The incidence of PEP in our study is 6.6%, which was better than that of previous studies. Liao et al. first demonstrated that a 5-minute EPBD reduces the risk of pancreatitis compared with the conventional 1-minute EPBD (4.8% vs. 15.1%, p = 0.038) [12]. Further systematic reviews showed that the duration of EPBD was inversely associated with pancreatitis risk [11]. Current clinical guideline suggests a EPBD duration of at least 2 minutes [7]. The mechanism of pancreatitis was thought to involve the compartment syndrome from the post-EPBD hemorrhage/edema of an uncut sphincter of Oddi [12, 17]. If the theory was true, the critical point of pancreatitis should be whether EPBD cause loosened sphincter ampullae, not the duration of the balloon dilatation. Our results showed that a dilatation time of at least 3 minutes with an adequate orifice opening did not increase the incidence of post-EPBD pancreatitis. Younger age was well-known as a risk factor of PEP [18], and post-EPBD compartment syndrome may explain why small CBD diameter was also an independent risk factor of pancreatitis.

The incidence of PEP in patients with EPLBD patients was 0.8–11.2% in previous studies [13, 14, 19]. However, none of the previous studies discussed the adequate duration of dilatation time in patients with EPLBD. The PEP incidence in patients with EPLBD was 1.9% in our study. The PEP rate were not significantly different among the three groups. All of our patients with EPLBD had their stones removed successfully, and EML was used in six of the patients (11.3%). Our results demonstrated that at least 3 minutes of dilatation duration may be enough for patients with EPLBD.

Recent studies have proven that prophylactic pancreatic stent placement had a significant reduction in incidence and severity of PEP, and pancreatic stent placement is recommended in patients who are at a high risk for PEP [20, 21]. In our study, no repeated inadvertent PD cannulation occurred, so we did not place any prophylactic stents in our patients. Non-steroidal anti-inflammatory drugs (NSAIDs) have also been shown to significantly reduce the incidence and severity of PEP [22–24]. Aggressive lactated Ringer's (LR) hydration is thought to prevent further injury to the pancreas from microvascular hypoperfusion and activate pancreatic enzymes [25]. Our data was collected before these promising studies, so rectal NSAIDs or LR solution were not administered to patients in our study for PEP prevention. However, our PEP incidence was not higher than current studies. Whether longer EPBD duration plus pancreatic stent placement, rectal NSAIDs, and LR solution hydration would further decrease the PEP incidence is interesting and warrants additional investigation.

With respect of successful stone removal, Liu et al. analyzed 10 randomized control trials (RCTs), and demonstrated that EPBD has equivalently great success for complete stone removal to EST [5]. Zhao et al. reviewed 14 RCTs and showed that EPBD decreased the overall clearance of stones compared to EST [10]. It is generally acknowledged that EPBD is less likely to remove large CBD stone. In our study, we did not exclude the large CBD stones, and the proportion of large CBD stone was about one third (CBD stone ≥ 10mm, n = 74, 37.4%). The successful stone removal rate was 100%, and the utility of EML was 3.5% (n = 7). Although there was a higher proportion of large CBD stones in patients in group C, the successful stone removal rate did not decrease. Our study showed that large CBD stones may increase the duration of EPBD, but did not decrease the clearance of CBD stones. The results were consistent with those reported by Liu et al.

Additionally, a previous meta-analysis showed that the post-EPBD bleeding rate was less than 1% [10]. The post-EPBD bleeding rate in our study was 1%, and there were no significant differences among the three groups. Furthermore, there were no procedure related perforations that occurred in our study, similar with that of a previous meta-analysis [5].

There were some limitations to our study. Our study was not a randomized control trial. There may be some selection bias in the study. The time of EPBD was not standardized but was the choice of the endoscopist, this may also cause bias. It is also not current standard not to use a guidewire for primary cannulation nowadays. However, the relatively low rate of PEP reflected the acceptable experience of the endoscopist performing biliary cannulation without using a guidewire at that time. In addition, our data were collected during the period when use of NSAIDs or LR was still debatable for PEP prevention, thus, only the patients from 2010 to 2012 when no rectal NSAIDs or LR solution hydration use in our hospital were included. It is interesting to investigate whether dilatation time could further affect the occurrence of PEP in the period when rectal NSAIDs was frequently used in high risk PEP patients. Further studies are still needed to determine the adequate balloon dilatation duration.

## Conclusions

An EPBD duration of more than 3 minutes with adequate orifice opened by an experienced endoscopist may be enough for PEP prevention. A small CBD diameter, young age and shorter dilatation duration could cause more PEP.

## Acknowledgments

We would like to acknowledge the assistance of Mr. Dong-Ming Liao.

## Author Contributions

**Formal analysis:** Chung-Kai Chou.

**Investigation:** Chung-Kai Chou, Kuei-Chuan Lee.

**Methodology:** Kuei-Chuan Lee.

**Resources:** Chung-Kai Chou, Kuei-Chuan Lee, Tseng-Shing Chen.

**Supervision:** Kuei-Chuan Lee, Jiing-Chyuan Luo, Tseng-Shing Chen, Chin-Lin Perng, Yi-Hsiang Huang, Han-Chien Lin, Ming-Chih Hou.

**Writing – original draft:** Chung-Kai Chou.

**Writing – review & editing:** Chung-Kai Chou, Kuei-Chuan Lee.

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
