## [Decision Letter · Decision Letter 0]

24 Mar 2020

PONE-D-19-32543

Endoscopic Papillary Balloon Dilatation less than Three Minutes for Biliary Stone Removal Increases the Risk of Post-ERCP Pancreatitis

PLOS ONE

Dear Dr. Lee,

Thank you for submitting your manuscript to PLOS ONE. After careful consideration, we feel that it has merit but does not fully meet PLOS ONE’s publication criteria as it currently stands. Therefore, we invite you to submit a revised version of the manuscript that addresses the points raised during the review process.

As you can see, your manuscript received somewhat mixed reviews. In particular, reviewer 2 pointed out important issues that need to be addressed such as (i) a limited size of the cohort; (ii) analysis of procedures, that were performed a relatively long time ago rather than focusing on a more recent cohort; (iii) issues with statistical analysis, that need to be addressed. Because of that, the expectations on the revised manuscript will be fairly high.

We would appreciate receiving your revised manuscript by May 08 2020 11:59PM. To enhance the reproducibility of your results, we recommend that if applicable you deposit your laboratory protocols in protocols.io, where a protocol can be assigned its own identifier (DOI) such that it can be cited independently in the future. For instructions see: http://journals.plos.org/plosone/s/submission-guidelines#loc-laboratory-protocols

We look forward to receiving your revised manuscript.

Kind regards,

Pavel Strnad

Academic Editor

PLOS ONE

Journal Requirements:

2. In the ethics statement in the manuscript, please provide additional information about the patient records used in your retrospective study. Specifically, please ensure that you have discussed whether all data were fully anonymized before you accessed them and/or whether the IRB or ethics committee waived the requirement for informed consent. If patients provided informed written consent to have data from their medical records used in research, please include this information.

Reviewers' comments:

Reviewer's Responses to Questions

**Comments to the Author**

1. Is the manuscript technically sound, and do the data support the conclusions?

Reviewer #1: Yes

Reviewer #2: Yes

2. Has the statistical analysis been performed appropriately and rigorously? 

Reviewer #1: Yes

Reviewer #2: I Don't Know

3. Have the authors made all data underlying the findings in their manuscript fully available?

Reviewer #1: Yes

Reviewer #2: No

4. Is the manuscript presented in an intelligible fashion and written in standard English?

Reviewer #1: Yes

Reviewer #2: Yes

5. Review Comments to the Author

Reviewer #1: 1. Can they give information on size of balloons used for papillary dilation (what was size in large balloon dilation).

2. Grammatical correction "systematic" instead of "systemic" in the discussion, 8th line

3. Information on complete stone removal on first attempt or repeat ERCP's were needed.

Reviewer #2: Kuei-Chuan Lee et al. present in their manuscript “Endoscopic Papillary Balloon Dilatation less than three minutes for biliary Stone Removal Increases the Risk of Post-ERCP-Pancreatitis” interesting data from a retrospective single centre observational study on 198 patients (11/2010-10/2012). The main finding was that in patients that underwent Endoscopic Papillary Balloon Dilatation (EPBD) for stone removal, dilatation duration < 3 minutes, lower CBD diameter and younger age were in a multivariate analysis independently associated with Post-ERCP-Pancreatitis (PEP).

As the authors have mentioned in their manuscript, a number of previous studies have already shown that the duration of EPBD was inversely associated with PEP; therefore, clinical guidelines already suggest an EPBD duration of at least 2 minutes.

The authors furthermore emphasize that no studies investigated so far, the impact of large balloon dilatation (EPLBD) on PEP. Nevertheless, the findings presented here are indeed not surprising but rather confirm previous studies.

It is unclear to me, why the authors choose the time intervals at 3min and 5min dilatation time to build the three groups that they compared? Why not 2min or 4 min? Was there an ROC-analysis performed to identify the optimal discriminatory time point?

I also wonder why only patients between 2010 and 2012 were included. For the endpoint PEP a long follow-up is not needed and therefore there is no obvoious reason not to include also patients from the years 2013-2018. This would also allow to include protective stenting and NSAIDs as protective factors.

Furthermore, in the Methods it is not mentioned which balloons and which sizes of balloons were used for EPBD and EPLBD. This essential information must be given.

Balloon Diameter (if different balloons were used) and also mechanical lithotripsy should additionally be included in the analysis of risk factors (table 4).

Minor limitations are the retrospective study design with a possible BIAS since the time of EPBD was not standardized but was the choice of the endoscopist.

It is also not current standard anymore not to use a guidewire for primary cannulation. However, the relatively low rate of PEP reflect the skills and experience of the endoscopist performing biliary cannulation without using a guidewire.

In Table 2 percentages should also be given for EPLBD.

An interesting but not significant finding was that no biliary tract infection was found in group A but 4.1% in group B and 8.5% in group C. If Group A would be compared with Group B and C together was the difference than statistically significant?

6. PLOS authors have the option to publish the peer review history of their article (what does this mean?). If published, this will include your full peer review and any attached files.

Reviewer #1: No

Reviewer #2: No

---

## [Author Response · Author response to Decision Letter 0]

21 Apr 2020

Reviewer #1: 

1. Can they give information on size of balloons used for papillary dilation (what was size in large balloon dilation).

Reply: Thanks for your comments. We added the information in the revised table2. EPLBD was defined as balloon size more than 10mm (revised table 2). 

2. Grammatical correction "systematic" instead of "systemic" in the discussion, 8th line

Reply: Thanks for your kind comment. We revised the grammatical error as your suggestion. (page 12, 1st line)

3. Information on complete stone removal on first attempt or repeat ERCP's were needed.

Reply: Thanks for your comments. Actually, all stones were removed successfully on first attempt and no repeat ERCP was done during the follow up period. The information was added in the revised table 2. 

Reviewer #2: 

Kuei-Chuan Lee et al. present in their manuscript “Endoscopic Papillary Balloon Dilatation less than three minutes for biliary Stone Removal Increases the Risk of Post-ERCP-Pancreatitis” interesting data from a retrospective single centre observational study on 198 patients (11/2010-10/2012). The main finding was that in patients that underwent Endoscopic Papillary Balloon Dilatation (EPBD) for stone removal, dilatation duration < 3 minutes, lower CBD diameter and younger age were in a multivariate analysis independently associated with Post-ERCP-Pancreatitis (PEP).

As the authors have mentioned in their manuscript, a number of previous studies have already shown that the duration of EPBD was inversely associated with PEP; therefore, clinical guidelines already suggest an EPBD duration of at least 2 minutes.

The authors furthermore emphasize that no studies investigated so far, the impact of large balloon dilatation (EPLBD) on PEP. Nevertheless, the findings presented here are indeed not surprising but rather confirm previous studies.

It is unclear to me, why the authors choose the time intervals at 3min and 5min dilatation time to build the three groups that they compared? Why not 2min or 4 min? Was there an ROC-analysis performed to identify the optimal discriminatory time point?

Reply: Thanks for your comments. The time point was not set by ROC analysis. However, a randomized trial by Liao et al. found that the risk of PEP was lower with a 5min EPBD as compared with those with a 1min EPBD (Gastrointest Endosc. 2010;72(6):1154-62). Therefore, we firstly chose 5min as a discriminatory point. Then we found that the mean of dilatation duration was 208.26±4.54 seconds, and the median was 210 seconds among the patients who received balloon dilatation duration less than 5 minutes (n= 127). For the convenience in daily practical use, we then chose 3min as another discriminatory point, which near the mean or median value in the patients receiving balloon dilatation time less than 5min. Finally, the time intervals at 3min and 5min were chosen. 

I also wonder why only patients between 2010 and 2012 were included. For the endpoint PEP a long follow-up is not needed and therefore there is no obvoious reason not to include also patients from the years 2013-2018. This would also allow to include protective stenting and NSAIDs as protective factors.

Reply: Thanks for your comments. We totally agree that protective stenting and rectal NSAIDs are important protective factors for PEP nowadays. However, during the period (2014-2016) when we retrospectively started to collect the data of patients receiving EPBD in our hospital, the use of rectal NSAIDs and stenting were still debatable. Therefore, we only chose the patients between 2010 and 2012 when no rectal NSAIDs or stenting was used in our ERCP room. Actually, we are interested in whether dilatation time could further affect the occurrence of PEP in another time period (2013-2018) when rectal NSAIDs was frequently used in high risk PEP patients in our hospital, future study will be initiated. (page 14, line 18)

Furthermore, in the Methods it is not mentioned which balloons and which sizes of balloons were used for EPBD and EPLBD. This essential information must be given.

Balloon Diameter (if different balloons were used) and also mechanical lithotripsy should additionally be included in the analysis of risk factors (table 4).

Reply: Thanks for your important suggestions. We had added information of balloons and size in the revised method (page 7, line 12) and Table 2. Mechanical lithotripsy had no significantly predictive value in univariate analysis (revised Table 4). 

Minor limitations are the retrospective study design with a possible BIAS since the time of EPBD was not standardized but was the choice of the endoscopist.

It is also not current standard anymore not to use a guidewire for primary cannulation. However, the relatively low rate of PEP reflect the skills and experience of the endoscopist performing biliary cannulation without using a guidewire.

Reply: Thanks for your comments. We added these comments as study limitations in the revised discussion (page 14, line 14).

In Table 2 percentages should also be given for EPLBD.

Reply: Thanks for your comment, we had revised that in the Table 2 as your suggestion. 

An interesting but not significant finding was that no biliary tract infection was found in group A but 4.1% in group B and 8.5% in group C. If Group A would be compared with Group B and C together was the difference than statistically significant?

Reply: We had analyzed that as your suggestion. However, there was no difference in biliary tract infection when compared the Group A (n=0) with the Groups B and C together (n=10) (p=0.170). (page 10, line 15).

---

## [Decision Letter · Decision Letter 1]

5 May 2020

Endoscopic Papillary Balloon Dilatation less than Three Minutes for Biliary Stone Removal Increases the Risk of Post-ERCP Pancreatitis

PONE-D-19-32543R1

Dear Dr. Lee,

We are pleased to inform you that your manuscript has been judged scientifically suitable for publication and will be formally accepted for publication once it complies with all outstanding technical requirements.

With kind regards,

Pavel Strnad

Academic Editor

PLOS ONE

Additional Editor Comments (optional):

Reviewers' comments:

Reviewer's Responses to Questions

**Comments to the Author**

1. If the authors have adequately addressed your comments raised in a previous round of review and you feel that this manuscript is now acceptable for publication, you may indicate that here to bypass the “Comments to the Author” section, enter your conflict of interest statement in the “Confidential to Editor” section, and submit your "Accept" recommendation.

Reviewer #2: All comments have been addressed

2. Is the manuscript technically sound, and do the data support the conclusions?

Reviewer #2: Yes

3. Has the statistical analysis been performed appropriately and rigorously? 

Reviewer #2: Yes

4. Have the authors made all data underlying the findings in their manuscript fully available?

Reviewer #2: Yes

5. Is the manuscript presented in an intelligible fashion and written in standard English?

Reviewer #2: Yes

6. Review Comments to the Author

Reviewer #2: In their revised manuscript Kuei-Chuan Lee et al. have addressed all the comments and suggestions from the reviewers. The manuscript has substantially improved and is therefore now acceptable for publication in PLOS One.

7. PLOS authors have the option to publish the peer review history of their article (what does this mean?). If published, this will include your full peer review and any attached files.

Reviewer #2: No

---

## [Editor Report · Acceptance letter]

15 May 2020

PONE-D-19-32543R1 

Endoscopic Papillary Balloon Dilatation less than Three Minutes for Biliary Stone Removal Increases the Risk of Post-ERCP Pancreatitis 

Dear Dr. Lee:

I am pleased to inform you that your manuscript has been deemed suitable for publication in PLOS ONE. Congratulations! Your manuscript is now with our production department. 

With kind regards,

on behalf of

Dr. Pavel Strnad 

Academic Editor

PLOS ONE